# Linear Bayesian Estimation of Misrecorded Poisson Distribution

**DOI:** 10.3390/e26010062

**Published:** 2024-01-11

**Authors:** Huiqing Gao, Zhanshou Chen, Fuxiao Li

**Affiliations:** 1School of Mathematics and Statistic, Qinghai Normal University, Xining 810008, China; huiqinggao0820@outlook.com; 2The State Key Laboratory of Tibetan Intelligent Information Processing and Application, Xining 810008, China; 3Faculty of Science, Xi’an University of Technology, Xi’an 710048, China; lifuxiao@xaut.edu.cn

**Keywords:** misrecorded Poisson distribution, linear Bayesian estimation, bootstrap

## Abstract

Parameter estimation is an important component of statistical inference, and how to improve the accuracy of parameter estimation is a key issue in research. This paper proposes a linear Bayesian estimation for estimating parameters in a misrecorded Poisson distribution. The linear Bayesian estimation method not only adopts prior information but also avoids the cumbersome calculation of posterior expectations. On the premise of ensuring the accuracy and stability of computational results, we derived the explicit solution of the linear Bayesian estimation. Its superiority was verified through numerical simulations and illustrative examples.

## 1. Introduction

Count data, which are typically characterized as data points that cannot be collected consecutively or are represented as 0 or natural numbers like 1, 2, 3, etc., are commonly encountered in various aspects of daily life. These data points are often effectively approximated by binomial or Poisson distributions. However, certain challenges may emerge during the actual counting process. When determining the number of defects per unit or item, there is a possibility that the recorder may inaccurately classify some units that actually contain one defect as perfect or defect-free while accurately recording other values. More specifically, some observations with a value of 1 are misclassified and reported as 0, while others, such as 2, 3, and 4, are correctly recorded. For this type of data containing erroneous records, direct modeling with a Poisson distribution is not appropriate. Instead, it can be approximated using a misrecorded Poisson distribution.

The misrecorded Poisson distribution, first proposed by Cohen [1], is often denoted as the Cohen–Poisson (CP) or MSR-Poisson distribution. Since then, many authors have studied this distribution and its various forms. For surveys, Dorris and Foote [2] provided a comprehensive survey and analysis of the impact of inspection errors on statistical quality control procedures. They discuss their effects on various aspects of statistical quality control procedures, including control charts, sample size, and inspection efficiency. Subsequently, the article introduces methods for estimating error probabilities and adjusting parameters in Poisson distributions, along with strategies for designing compensatory plans to mitigate the impact of inspection errors. All these efforts are grounded in Cohen’s seminal work, which developed multiple estimators for error probability assessment and parameter recalibration in Poisson distributions, thereby establishing a significant theoretical framework. Gupta et al. [3] studied when the observed frequency of zeros is significantly higher or lower than predicted by the model; the adjusted random variable can be described as a mixture of two Poisson distributions. This distribution can be regarded as a mixture of two distributions, one of which is degenerate at zero. Zhang et al. [4] study the zero-and-one inflated Poisson (ZOIP) distribution, develop likelihood-based inference methods, and provide simulations and real data examples to illustrate the proposed methods. The key properties of the ZOIP distribution are established, including five equivalent stochastic representations and other important distributional properties. Maximum likelihood estimates of parameters are obtained through Fisher scoring and the expectation–maximization algorithm, and bootstrap confidence intervals and hypothesis testing methods are provided for parameters in large samples. Liu [5] et al. introduce a new multivariate ZAP distribution, building on the traditional multivariate Poisson distribution. A key feature of this distribution is its ability to model count vectors that are zero-truncated, zero-deflated, or zero-inflated with a more flexible dependency structure. This means that correlation coefficients between components can be both positive and negative. They introduce an expectation–maximization (EM) algorithm for calculating the maximum likelihood estimates (MLEs) and posterior modes of parameters. Bagui and Mehra [6] gave a historical background for the Poisson distribution and also described some of its applications in the early days and more. Additionally, they introduced a new ratio method that demonstrates convergence to the normal distribution, which may attract attention. In the context of Poisson processes, considering the swift progress in current industrial and technological spheres, statistical models must advance to keep up with the times, adapting to handle data with higher frequencies and greater complexity.

The Hawkes process model, noted for its self-excitation and time-dependence, serves as an extension to the Poisson process. This extension offers more sophisticated and adaptable tools for analyzing and forecasting error patterns in ever-changing environments. For relevant research results, refer to Zhang et al. [7] and Wang and Zhang [8]. Lamprinakou et al. [9] proposed a new epidemiological model using the Hawkes process to study the spread of COVID-19. It focuses on estimating unobserved infection cases, offering insights into the disease’s transmission dynamics.

Furthermore, numerous monographs and doctoral dissertations have investigated the misrecorded Poisson distribution and its various forms. Xu [10] studied the statistical properties of a Poisson generalized inverse Gaussian distribution, a negative binomial distribution, a Poisson inverse Gaussian distribution, and a Poisson inverse gamma distribution. The model they studied is similar to the model studied in this article, which is very helpful for the study of properties and parameter derivation methods in this article. Johnson [11] has effectively synthesized the findings from various studies on the misrecorded Poisson distribution and its parameter estimation, encompassing research conducted by Cohen as well as contributions from other researchers in the field. Djuraš [12] discussed the one-displaced misrecorded Poisson and size-biased misrecorded Poisson distributions, deriving their parameter estimators. Tuwei [13] derived the probability density function from the probability generating function and provided expressions for the mean, the variance, and the relationship between parameters based on this probability density function.

While the majority of previous studies have focused on classical estimation methods, Angers and Biswas [14] employ Bayesian methods for parameter estimation and prediction. Specifically, they contemplate a zero-inflated generalized Poisson model, which encompasses three parameters: the zero-inflation parameter, the dispersion parameter of the generalized Poisson distribution, and the mean parameter. For the purpose of Bayesian estimation, the article utilizes appropriate prior distributions and employs a Monte Carlo integration technique via importance sampling to obtain the posterior distributions. Subsequently, the expected values of the posterior distributions are computed, serving as the Bayesian estimates of the parameters. Rodrigues [15] studied the zero-inflated distributions from a Bayesian point of view using a data augmentation algorithm. Wang [16] considered a 0–1 expansion Poisson regression model with covariates and proposed a Bayesian estimation of the model parameters.

Compared to traditional Bayesian estimation methods, it is worth noting that the linear Bayesian method, initially proposed by Hartigan [17], has gained popularity as a simple Bayesian approach for parameter estimation. Rao [18] studied the linear Bayesian method from the perspective of linear optimization and argued that this method better considers the prior uncertainty and uses this “incomplete” information to construct linear Bayesian parameter estimates. LaMotte [19] introduced a linear Bayesian estimator that has the smallest total mean square error among all linear estimators. Hesselager [20] demonstrated that if the average risk of an empirical linear Bayesian estimate converges to the risk of the corresponding linear Bayesian estimate, then it is asymptotically optimal in the usual sense. Goldstein [21] adeptly modified the linear Bayesian estimator, specifically designed for estimating the mean of a distribution whose form is unknown, by incorporating the use of an estimate derived from sample variance. Hoffmann [22] applied the linear Bayesian method to estimate an unknown parameter vector in the linear regression model with ellipsoidal parameter constraints. The conditions under which certain linear empirical Bayesian estimators are superior to the standard estimator for an arbitrary k≥1 are given by Samaniego and Vestrup [23]. Wei and Zhang [24] verified under the Predictive Pitman Closeness criterion and Posterior Pitman Closeness that the linear Bayesian method outperforms the generalized least squares method for linear models. More recently, Lin [25] employed Monte Carlo simulations and empirical computations to compare the outcomes of constrained least-squares estimation and constrained linear Bayesian estimation, as well as their distances from the Bayesian estimates. This approach serves to validate the superiority of constrained linear Bayesian estimation over constrained least-squares estimation. Tao [26] provided an empirical linear Bayesian approach, conducting an empirical analysis using a Bayesian model for multiple insurance contracts with Pareto distributions. This method does not rely on any prior distribution information. Liu [27] studied the parameter estimation problem of singular linear models with equality constraints and concluded that the higher the degree of singularity of the model, the more obvious the superiority of the best homogeneous linear Bayesian unbiased estimation over the least-squares estimation. Chen [28] introduces a linear Bayesian method for estimating parameters in extreme value distributions from Type II censored samples. This approach, which combines Bayesian and optimization techniques, offers a straightforward and practical solution, overcoming the complexity often associated with classical Bayesian estimation. The method’s advantages, particularly for small or heavily censored samples, are validated through numerical experiments, demonstrating its effectiveness when compared to maximum likelihood and unbiased estimations. The models, parameter estimation methods, and criteria for assessing their effectiveness used in the aforementioned work have significantly contributed to the research presented in this paper. These models, encompassing a range of statistical, probabilistic, and computational approaches, form the fundamental basis of our analysis. They provide a structured and systematic framework that aids in the coherent understanding and interpretation of the complex data sets we are dealing with. The selection of a particular model is heavily dependent on the specific nature of the data under study and the overarching research question, with each model introducing its own set of assumptions and perspectives to the analysis.

Parameter estimation methods are a key component in refining and fine-tuning these models. These methods are instrumental in extracting significant insights from the data, as they focus on determining the values of the model parameters that most accurately represent the observed data. Depending on the model’s nature and the data’s characteristics, various techniques like maximum likelihood estimation, Bayesian inference, or least-squares fitting are utilized. The precision and reliability of these methods are paramount, as they have a direct bearing on the validity and credibility of our research findings. Therefore, the judicious selection and application of these methods are critical.

Given the notable advantages of linear Bayesian estimators, this paper specifically focuses on parameter estimation for the misrecorded Poisson distribution using linear functions derived from sample data. We employ the criterion of minimizing the mean square error to formulate linear Bayesian estimation expressions for the distribution’s parameters. To our knowledge, no existing studies have explored linear Bayesian estimation specifically for parameters within the misrecorded Poisson distribution. Traditional methods often fall short in addressing the unique challenges posed by misrecorded data, such as biased estimates or increased error variance. Therefore, our method fills a significant gap in the statistical methodology by providing a tailored solution for this specific distribution. Furthermore, we use the criterion of minimizing the mean square error to formulate linear Bayesian estimation expressions for the distribution’s parameters. This criterion ensures that the estimations are not only accurate but also consistent by minimizing the average of the squares of the errors—the difference between the estimator and what is estimated. This approach optimizes the balance between bias and variance in the estimations, leading to more reliable results.

Our work stands out not only for its innovative approach to handling the misrecorded Poisson distribution but also for its potential wide-ranging applications in various fields. In areas where misrecorded count data are common, such as epidemiology, environmental studies, and quality control in manufacturing, the implications of this research are substantial. By introducing new insights and techniques for managing such distributions, our study promises to enhance the accuracy and reliability of data analysis in these fields, leading to more informed and dependable conclusions. This could mark a significant advancement in statistical methods, opening doors to more robust and sophisticated data analysis techniques in these critical areas of study.

The rest of this paper is structured as follows: Section 2 offers a detailed examination of the misrecorded Poisson distribution and delves into its maximum likelihood estimation. In Section 3, we introduce and elaborate on a linear Bayesian estimation method, highlighting its superior estimation results, primarily based on the principle of the mean square error matrix. Section 4 presents numerical simulations and example analyses. Lastly, in Section 5, we summarize the entire study, provide our conclusions, point out the shortcomings of the article, and suggest future research directions

## 2. Misrecorded Poisson Distribution

The probability mass function of the misrecorded Poisson distribution is
(1)$$P(X=x)=\left\{ \begin{array}{l} e^{-\lambda }(1+\lambda \phi )x=0\\ e^{-\lambda }\lambda (1-\phi )x=1\\ e^{-\lambda }\frac{\lambda^{x}}{x!}\;x=2,3,\ldots \end{array}\right.,$$
where λ>0, 0<ϕ<1. ϕ is the probability of misclassification or the proportion of ones that are reported as zeros. The misrecorded Poisson distribution is reduced to a Poisson distribution when ϕ=0. Also, all the observations falling in class one will be reported as zero when ϕ=1; thus, it is a zero–one modified distribution. We can say that the misrecorded Poisson distribution is zero-inflated (there is an excess of zeros) and one-deflated (there are fewer ones than expected). If $\phi_{0}=\lambda \phi e^{-\lambda }>0$, then the Poisson’s zero probability is increased (zero inflation), while the Poisson’s one probability is decreased for $\phi_{1}=-\lambda \phi e^{-\lambda }<0$.

The corresponding probability generating function, the mean, the variance, and the second, third, and fourth moments are given as, respectively,
$$G_{X}(s)=e^{-\lambda }(1+\lambda \phi )+\lambda e^{-\lambda }(1-\phi )s+\sum_{x=2}^{\infty }\frac{{\left(\lambda s\right)}^{x}e^{-\lambda }}{x!}$$
$$=e^{-\lambda }(1+\lambda \phi )+\lambda se^{-\lambda }(1-\phi )+\left\{e^{-\lambda (s-1)}-\lambda se^{-\lambda }-e^{-\lambda }\right\}$$
$$=e^{-\lambda }+\lambda \phi e^{-\lambda }+\lambda se^{-\lambda }-\lambda \phi se^{-\lambda }+e^{-\lambda (s-1)}-\lambda se^{-\lambda }-e^{-\lambda }$$
$$=\lambda \phi e^{-\lambda }-\lambda \phi se^{-\lambda }+e^{-\lambda (s-1)},$$
$$E(X)=\mu =\lambda (1-\phi e^{-\lambda }),$$
$$Var(X)=C_{2}(0)=\sigma_{X}^{2}=\lambda^{2}+(1+\phi e^{-\lambda })(1-\lambda (\phi e^{-\lambda })),$$
$$E(X^{2})=\mu_{\left(0\right)}=\lambda^{2}+\lambda (1-\phi e^{-\lambda }),$$
$$E(X^{3})=\mu_{\left(0,0\right)}=\lambda^{3}+3\lambda^{2}+(1-\phi e^{-\lambda }),$$
$$E(X^{4})=\mu_{\left(0,0,0\right)}=\lambda^{4}+6\lambda^{3}+7\lambda^{2}+(1-\phi e^{-\lambda }).$$
assuming that the random variables $x_{1},x_{2},\ldots \ldots ,x_{N}$ are samples consisting of N observations from the misrecorded Poisson distribution, with $n_{0}$ denoting the number of zero observations and $n_{1}$ denoting the number of one observations. The likelihood function is
$$L(x_{1},x_{2},\cdot \cdot \cdot x_{N};\lambda ,\phi )={\left[e^{-\lambda }(1+\phi \lambda )\right]}^{n_{0}}{\left[(1-\phi )\lambda e^{-\lambda }\right]}^{n_{1}}\prod *e^{-\lambda }\lambda^{x_{i}}/x_{i}$$
(2)$$=e^{-N\lambda }{\left(1+\phi \lambda \right)}^{n_{0}}{\left(1-\phi \right)}^{n_{1}}\lambda^{\sum_{i=1}^{N}x_{i}}{\left[\prod *x_{i}!\right]}^{-1},$$
where ∏* is the product of $x_{1},x_{2},\ldots \ldots ,x_{N}$ that are neither zero nor one. Taking the logarithms of Equation (2) and letting its partial derivatives with respect to λ and ϕ, respectively, equal zero, we obtain the following equation:$$\partial \log L/\partial \lambda =-N+n_{0}\phi /(1+\phi \lambda )+\sum_{i=1}^{N}x_{i}/\lambda =0.$$
$$\partial \log L/\partial \phi =n_{0}\lambda (1+\phi \lambda )-n_{1}/\left(1-\phi \right)=0,$$
and then we have:(3)$$\lambda^{2}-(\bar{x}-1+n_{0}/N)\lambda -(\bar{x}-n_{1}/N)=0,$$
(4)$$\phi =[n_{0}-n_{1}/\lambda ]/(n_{0}+n_{1}),$$
where $\bar{x}=\sum_{i=1}^{N}x_{i}/N$ is the mean of the sample. Solving the above equation yields the maximum likelihood estimates ${\hat{\lambda }}_{MLE}$ and ${\hat{\phi }}_{MLE}$ for λ and ϕ:(5)$${\hat{\lambda }}_{MLE}=\left[\left(\bar{x}-1+n_{0}/N\right)+\sqrt{(\bar{x}-1+{n_{0}/N)}^{2}+4(\bar{x}-n_{1}/N)}\right]/2,$$
(6)$${\hat{\phi }}_{MLE}=\left(n_{0}-n_{1}/{\hat{\lambda }}_{MLE}\right)/\left(n_{0}+n_{1}\right).$$ Translating the solution of this estimate into matrix form to express it, we have
(7)$${\hat{\theta }}_{MLE}=({\hat{\lambda }}_{MLE},{\hat{\phi }}_{MLE}{)}^{\prime }=\left(\begin{array}{cc} 1/2 & 0\\ 0 & \frac{1}{n_{0}+n_{1}} \end{array}\right)\left(\begin{array}{c} t_{1}\\ t_{2} \end{array}\right)≜AT,$$
where
(8)$$A=\left(\begin{array}{cc} 1/2 & 0\\ 0 & \frac{1}{n_{0}+n_{1}} \end{array}\right),T=\left(\begin{array}{c} t_{1}\\ t_{2} \end{array}\right).$$

## 3. Linear Bayesian Estimation

### 3.1. The Expressions of Linear Bayesian Estimation

**Definition** **1.***Assume that the prior distribution function,* π(θ) *of* $\theta =(\lambda ,\phi {)}^{\prime }$*, satisfies the family of prior distributions:*(9)$$\xi =\left\{\pi (\theta ):E{\left\|\theta \right\|}^{2}<\infty \right\}.$$

Let ${\hat{\theta }}_{LB}=BT+b$ be a linear estimate of the parameter θ under the statistic T, in which B and b are 2×2- and 2×1-dimensional unknown matrices, respectively. Then, ${\hat{\theta }}_{LB}$ is called the linear Bayesian estimate of θ if ${\hat{\theta }}_{LB}$ satisfies the following conditions:(10)$$E_{\left(T,\theta \right)}({\hat{\theta }}_{LB}-\theta )=0,$$
(11)$$R({\hat{\theta }}_{LB},\theta )=\underset{B,b}{\min}E_{\left(T,\theta \right)}L(\hat{\theta }-\theta ),$$
where $E_{\left(T,\theta \right)}$ denotes the expectation of the joint distribution of T and θ.

The function
(12)$$L({\hat{\theta }}_{LB},\theta )=({\hat{\theta }}_{LB},\theta {)}^{\prime }D({\hat{\theta }}_{LB},\theta ),$$
where D is a 2×2 positive definite matrix, is called the squared loss function. Minimizing the squared loss, we can obtain the expressions of B and b, which are given in the following theorem.

**Theorem** **1.***Suppose that the prior distribution,* π(θ)*, of the parameter vector* $\theta =(\lambda ,\varphi {)}^{\prime }$ *in the misrecorded Poisson distribution (1) satisfies the family of prior distributions (9). Then, under the squared loss function, the linear Bayesian estimate,* ${\hat{\theta }}_{LB}$*, of* θ *is*(13)$${\hat{\theta }}_{LB}=\left(A-AWM\right)T+AWMA^{-1}E\theta ,$$(14)b=(I−B)E(θ)=(I−B)μ,*where*(15)W=E[Cov(T,θ)],(16)$$M={\left[W+A^{-1}Cov(\theta )(A^{-1}{)}^{\prime }\right]}^{-1}.$$

**Proof.**  According to the condition in Equation (10), we have
$$0=E[E({\hat{\theta }}_{LB}-\theta |\theta )]=E\left[E(BT+b)-\theta \right],$$
so
$$BE_{\left(T,\theta \right)}(T)+b=E\theta ,$$
(17)$$b=E\theta -BA^{-1}E\theta .$$According to the condition in Equation (11), we have
$$E_{\left(T,\theta \right)}L(\hat{\theta },\theta )=E_{\left(Y,\theta \right)}\{[BT+b-\theta {]}^{\prime }D[BT+b-\theta ]\}$$
$$=E_{\left(T,\theta \right)}L(\hat{\theta },\theta )$$
$$=E_{\left(T,\theta \right)}\{[BT+E\theta -BA^{-1}E\theta -\theta {]}^{\prime }D[BT+E\theta -BA^{-1}E\theta -\theta ]\}.$$
$$E_{\left(T,\theta \right)}\{tr(D[BT+E\theta -BA^{-1}E\theta -\theta ][BT+E\theta -BA^{-1}E\theta -\theta {]}^{\prime })\}$$
$$=E_{\left(T,\theta \right)}\{tr(D[B(T-A^{-1}E\theta )-(\theta -E\theta )][B(T-A^{-1}E\theta )-(\theta -E\theta ){]}^{\prime })\}$$
$$=tr(DE_{\left(T,\theta \right)}\{[B(T-A^{-1}E\theta )-(\theta -E\theta )][B(T-A^{-1}E\theta )-(\theta -E\theta ){]}^{\prime }\})$$
$$=tr(DE_{\left(T,\theta \right)}\{B(T-A^{-1}E\theta )(T-A^{-1}E\theta {)}^{\prime }B^{\prime }-B(T-A^{-1}E\theta )(\theta -E\theta {)}^{\prime }$$
$$-(\theta -E\theta )(T-A^{-1}E\theta {)}^{\prime }B^{\prime }+(\theta -E\theta )(\theta -E\theta {)}^{\prime }\})$$
$$=tr(BDE_{\left(T,\theta \right)}[(T-A^{-1}E\theta )(T-A^{-1}E\theta {)}^{\prime }]B^{\prime })-tr(DBA^{-1}Cov(\theta ))$$
(18)$$-tr(DCov(\theta )(A^{-1}{)}^{\prime }B^{\prime })+tr(DCov(\theta )).$$In the above equation,
$$E_{\left(T,\theta \right)}[(T-A^{-1}E\theta )(T-A^{-1}E\theta {)}^{\prime }]$$
$$=E_{\left(T,\theta \right)}[TT^{\prime }-T(A^{-1}E\theta {)}^{\prime }-A^{-1}E\theta T^{\prime }+A^{-1}E\theta (A^{-1}E\theta {)}^{\prime }]$$
$$=E_{\left(T,\theta \right)}TT^{\prime }+A^{-1}E\theta (A^{-1}E\theta {)}^{\prime }-E_{\left(T,\theta \right)}T(A^{-1}E\theta {)}^{\prime }-(A^{-1}E\theta )E_{\left(T,\theta \right)}T^{\prime }]$$
$$=E_{\left(T,\theta \right)}TT^{\prime }+A^{-1}E\theta (A^{-1}E\theta {)}^{\prime }-A^{-1}E\theta (A^{-1}E\theta {)}^{\prime }-A^{-1}E\theta (A^{-1}E\theta {)}^{\prime }$$
$$=E_{\left(T,\theta \right)}TT^{\prime }-A^{-1}E\theta (A^{-1}E\theta {)}^{\prime }$$
$$=E[E(TT^{\prime }|\theta )]-A^{-1}E\theta (A^{-1}E\theta {)}^{\prime }$$
$$=E\{E[(T-A^{-1}\theta +A^{-1}\theta )(T-A^{-1}\theta +A^{-1}\theta {)}^{\prime }|\theta ]\}-A^{-1}E\theta (A^{-1}E\theta {)}^{\prime }$$
$$=E\{E[A^{-1}\theta (A^{-1}\theta {)}^{\prime }+(T-A^{-1}\theta )(T-A^{-1}\theta {)}^{\prime }+(T-A^{-1}\theta )(A^{-1}\theta {)}^{\prime }$$
$$+(A^{-1}\theta )(T-A^{-1}\theta {)}^{\prime }|\theta ]\}-A^{-1}E\theta (A^{-1}E\theta {)}^{\prime }$$
$$=E\{A^{-1}\theta (A^{-1}\theta {)}^{\prime }+E[(T-A^{-1}\theta )(T-A^{-1}\theta {)}^{\prime }]+E[(T-A^{-1}\theta )(A^{-1}\theta {)}^{\prime }]$$
$$=E[A^{-1}\theta (A^{-1}\theta {)}^{\prime }+E((T-A^{-1}\theta )(T-A^{-1}\theta {)}^{\prime })]-A^{-1}E\theta (A^{-1}E\theta {)}^{\prime }$$
$$+E(A^{-1}\theta (A^{-1}\theta {)}^{\prime })\}-A^{-1}E\theta (A^{-1}E\theta {)}^{\prime }$$
$$=E[A^{-1}\theta (A^{-1}\theta {)}^{\prime }]+E(Cov(T|\theta ))-A^{-1}E\theta (A^{-1}E\theta {)}^{\prime }$$
$$=E(Cov(T|\theta ))+A^{-1}[E(\theta \theta^{\prime })-E(\theta )E(\theta {)}^{\prime }](A^{-1})$$
$$=W+A^{-1}Cov(\theta )(A^{-1}{)}^{\prime }.$$Therefore,
$$E_{\left(T,\theta \right)}L(\hat{\theta },\theta )=tr(DB(W+A^{-1}Cov(\theta )(A^{-1}{)}^{\prime })B^{\prime })-tr(DBA^{-1}Cov(\theta ))$$
$$-tr(DCov(\theta )(A^{-1}{)}^{\prime }B^{\prime })+tr(DCov(\theta )).$$By taking a partial derivative of the matrix B in the above equation, we have
$$\frac{\partial R(\hat{\theta },\theta )}{\partial B}=2DB[W+A^{-1}Cov(\theta )(A^{-1}{)}^{\prime }]-2DCov(\theta )(A^{-1})=0,$$
(19)$$B=A-AW{\left[W+A^{-1}Cov(\theta )(A^{-1}{)}^{\prime }\right]}^{-1}=A-AWM,$$
where
W=E[Cov(T,θ)]
$$=E[(T-A^{-1}\theta )(T-A^{-1}\theta {)}^{\prime }|\theta ]$$
$$=E\left\{E\left[\left(\begin{array}{c} t_{1}\\ t_{2} \end{array}\right)-\left(\begin{array}{cc} 2 & 0\\ 0 & \frac{1}{n_{0}+n_{1}} \end{array}\right)\left(\begin{array}{c} \lambda \\ \phi \end{array}\right)\right]{\left[\left(\begin{array}{c} t_{1}\\ t_{2} \end{array}\right)-\left(\begin{array}{cc} 2 & 0\\ 0 & \frac{1}{n_{0}+n_{1}} \end{array}\right)\left(\begin{array}{c} \lambda \\ \phi \end{array}\right)\right]}^{\prime }|\theta \right\}$$
$$=\left[\begin{array}{cc} E{\left(t_{1}-2\lambda \right)}^{2} & 0\\ 0 & E{\left(t_{2}-(n_{0}+n_{1})\lambda \right)}^{2} \end{array}\right]$$
(20)$$=\left[\begin{array}{cc} Var(t_{1}) & 0\\ 0 & Var(t_{2}) \end{array}\right].$$Theorem 1 proved. □

An expression for linear Bayesian estimation, as detailed in the previously stated Theorem 1, has been constructed. This expression is derived while adhering to the constraints of unbiasedness and focused on the minimization of risk.

### 3.2. The Superiority of Linear Bayesian Estimation

In this subsection, we focus on examining the superiority of linear Bayesian estimation. To evaluate the effectiveness of this estimator, we utilize the mean square error (MSE), a well-regarded measure for assessing how closely an estimator approximates the actual parameter it seeks to estimate. Additionally, for a more careful comparison, we also consider the mean square error matrix (MSEM). To assess the superiority of ${\hat{\theta }}_{LB}$ according to the MSEM criterion, let us start by defining the MSEM.

**Definition** ** 2.***Let* $\hat{\theta }$ *be an estimator of the parameter vector* θ*. The MSE of* $\hat{\theta }$ *is defined as*(21)$$MSE(\theta )=E[(\hat{\theta }-\theta {)}^{\prime }(\hat{\theta }-\theta )],$$ *and the MSEM of* 
$\hat{\theta }$ *is defined as*
(22)$$MSEM(\theta )=E[(\hat{\theta }-\theta )(\hat{\theta }-\theta {)}^{\prime }].$$

Let the parameter vector θ be estimated in two different ways as ${\hat{\theta }}_{1}$ and ${\hat{\theta }}_{2}$. If
$$MSEM({\hat{\theta }}_{2})-MSEM({\hat{\theta }}_{1})\geq 0,$$
or
(23)$$(MSEM({\hat{\theta }}_{2})-MSEM({\hat{\theta }}_{1})\geq 0),$$
then ${\hat{\theta }}_{1}$ is said to be better than ${\hat{\theta }}_{2}$ under the MSEM or MSE criterion, and it is clear that the MSEM criterion is stronger than the MSE criterion.

From the previous section, the maximum likelihood estimation, ${\hat{\theta }}_{MLE}$, can be expressed as the product of the statistic T and the matrix A, and the linear Bayesian estimation, ${\hat{\theta }}_{LB}$, is also constructed from the statistic T. Therefore, in this section, the MSEM criterion is applied to compare ${\hat{\theta }}_{LB}$ and ${\hat{\theta }}_{MLE}$.

**Theorem** **2.***The linear Bayesian estimation,* ${\hat{\theta }}_{LB}$*, defined in (7) is preferred to the maximum likelihood estimation,*  ${\hat{\theta }}_{MLE}$*, defined in (13) under the MSEM criterion, i.e.,*
$$MSEM({\hat{\theta }}_{LB})\leq MSEM({\hat{\theta }}_{MLE})$$

**Proof.** 

$$MSEM({\hat{\theta }}_{LB})=E_{\left(T,\theta \right)}[({\hat{\theta }}_{LB}-\theta )({\hat{\theta }}_{LB}-\theta {)}^{\prime }]$$


$$=E[Cov({\hat{\theta }}_{LB}-\theta |\theta )]+Cov[E({\hat{\theta }}_{LB}-\theta |\theta )]$$


$$=E[Cov({\hat{\theta }}_{LB}|\theta )]+Cov[E({\hat{\theta }}_{LB}-\theta |\theta )]$$


$$=E[Cov(A-AWM)T+AWMA^{-1}E\theta |\theta )]$$


$$+Cov[E(AT-AWM(T-A^{-1}E\theta )-\theta |\theta )]$$


$$=(A-AWM)E(Cov(T|\theta )(A-AWM{)}^{\prime })$$


$$+Cov[E(AT|\theta )-E(AWM(T-A^{-1}E\theta |\theta )-\theta )]$$


$$=(A-AWM)W(A-AWM{)}^{\prime }+Cov(-AWME(T-A^{-1}E\theta |\theta ))$$


$$=(A-AWM)W(A-AWM{)}^{\prime }+Cov(AWMA^{-1}E[AT-E\theta |\theta ])$$


$$=A[(I-WM)W(I-WM)+WMA^{-1}Cov(\theta )(A^{-1}{)}^{\prime }MW]A^{\prime }$$


$$=A[W-2WMW+WM(W+A^{-1}Cov(\theta )(A^{-1}{)}^{\prime })MW]A^{\prime }$$


$$=A[W-WMW]A^{\prime }.$$

However,
$$MSEM({\hat{\theta }}_{MLE})=E_{\left(T,\theta \right)}[(\hat{\theta }-\theta )(\hat{\theta }-\theta {)}^{\prime }]$$
$$=E\{E[(\hat{\theta }-\theta )(\hat{\theta }-\theta {)}^{\prime }|\theta ]\}$$
$$=E\{E[(AT-\theta )(AT-\theta {)}^{\prime }|\theta ]\}$$
$$=AWA^{\prime }.$$According to the defining equation of W, we can easily find that W is positive and definite, and because of the definition of the matrix M, both $A^{-1}$ and $(A^{-1}{)}^{\prime }$ are positive definite matrices. Thus, Theorem 2 is proved. *□*

## 4. Numerical Simulation and Empirical Application

### 4.1. Numerical Simulation

In this section, we evaluate the performance superiority of classical and linear Bayesian estimation methods for the misrecorded Poisson distribution using numerical simulations. By choosing a range of different parameter values, we compute the mean square error (MSE) for both classical and linear Bayesian estimations relative to these values. A smaller error indicates a more accurate estimation. The procedures for these numerical simulations are detailed in the following steps.

1. Generate a dataset that accurately follows the characteristics of the misrecorded Poisson distribution, ensuring to use the predefined true parameter values as a foundational reference.

2. Proceed to calculate and derive parameter estimates, utilizing and comparing different established estimation methods for the analysis.

3. Calculate the mean and variance of the statistic T defined in Equation (20) using the bootstrap sampling method, and then use these values in the computation.

4. Repeat the above steps J times to obtain the mean of the estimate and give the MSE.

We set the sample size to N = 200 and 800 and set the parameters in the misrecorded Poisson distribution (1) to very between λ=0.7 and ϕ=0.7; λ=0.7 and ϕ=0.9;λ=0.9 and ϕ=0.7; and λ=0.9 and ϕ=0.9. All the simulations are performed via J=100 repetitions. The results are listed in Table 1 and Table 2. In these two tables, ${\hat{\lambda }}_{LB}$ and ${\hat{\phi }}_{LB}$ represent the linear Bayesian estimates of λ and ϕ, ${\hat{\theta }}_{MLEMSE}$ denotes the MSE for the maximum likelihood estimation, ${\hat{\theta }}_{LBMSE}$ denotes the MSE of the linear Bayesian estimation, and ${\left\|{\hat{\theta }}_{MLEMSE}-{\hat{\theta }}_{LBMSE}\right\|}_{2}$ is the Euclidean distance between ${\hat{\theta }}_{MLEMSE}$ and ${\hat{\theta }}_{LBMSE}$.

From Table 1, we can see that all the estimates are close to the true values regardless of the values of λ and ϕ. Importantly, our analysis reveals a key trend: as the sample size increases, all the estimates progressively converge towards their true values. This pattern is evident for both the maximum likelihood estimation and the linear Bayesian estimation, underscoring their consistency as estimators. The reliability of these approaches is evidenced in Table 2, which presents a noteworthy distinction in the mean square error (MSE) values between them. Specifically, the MSE for the linear Bayesian estimation is lower than that for maximum likelihood estimation. Remarkably, the Euclidean distance between ${\hat{\theta }}_{MLEMSE}$ and ${\hat{\theta }}_{LBMSE}$ diminishes as the sample size increases, suggesting that the advantage of linear Bayesian estimation becomes more pronounced in smaller sample sizes.

### 4.2. Empirical Application

In this section, we illustrate our proposed method using two sets of real data.

#### 4.2.1. Empirical Application 1

The first dataset for our research originates from Ladislaus Bortkiewicz’s acclaimed book ‘*Das Gesetz der Kleinen Zahlen*’, published in 1898 [29]. This particular data set gained historical significance when Cohen introduced the misrecorded Poisson distribution model for the first time. It comprises a collection of statistical records detailing the annual fatalities among Prussian soldiers due to horse-related incidents from 1875 to 1884. The data encompass detailed records from 14 distinct regiments and one guard unit, providing a diverse and comprehensive dataset for the analysis. The nature of these incidents inherently suggests the suitability of a Poisson distribution for modeling the frequency and patterns of these rare events. In an effort to validate and corroborate the findings discussed in this paper, we undertook some modifications to the original dataset. Specifically, we adjusted 20 entries that were originally noted as having a value of 1, revising them to 0. This adjustment is carefully documented in Table 3, allowing for a more accurate analysis in line with our study objectives.

We now fit a misrecorded Poisson distribution to this modified dataset using both maximum likelihood estimation and linear Bayesian estimation. From Table 3, we have N=200, $n_{0}=129$, $n_{1}=45$, $\bar{x}=102/200=0.51$, $n_{0}/N=0.645$, and $n_{1}/N=0.225$. Substituting these values into the maximum likelihood estimation, as shown in Equations (5) and (6), we have
$${\hat{\lambda }}_{MLE}=\left[\left(\bar{x}-1+n_{0}/N\right)+\sqrt{(\bar{x}-1+{n_{0}/N)}^{2}+4(\bar{x}-n_{1}/N)}\right]/2$$
$$=\left[\left(0.51-1+0.645\right)+\sqrt{{\left(0.51-1+0.645\right)}^{2}+4\times (0.51-0.225)}\right]/2$$
=0.61695.
$${\hat{\phi }}_{MLE}=\left(n_{0}-n_{1}/\hat{\lambda }\right)/\left(n_{0}+n_{1}\right)$$
$$=\left((129-45)/\hat{\lambda }\right)/\left(129+45\right)$$
=0.322187.

From the unchanged raw data [29], we have that λ=0.610 and φ=20/65=0.308, which is the proportion of ones that were misclassified in the process of altering the original data for this illustration. The specific proof process can be found in Appendix A. Utilizing linear Bayesian estimation (13), we obtain ${\hat{\lambda }}_{LB}=0.6167831$ and ${\hat{\varphi }}_{LB}=0.3221758$. Subsequently, we can compute the MSE by comparing the true values of the two parameters with the combined estimations from above: $MSE({\hat{\theta }}_{MLE})=0.0001247861$ and $MSE({\hat{\theta }}_{LB})=$ 0.0001235136, so $MSE({\hat{\theta }}_{LB})\leq MSE({\hat{\theta }}_{MLE})$. This indicates that the linear Bayesian estimation has better performance than the maximum likelihood estimation.

#### 4.2.2. Empirical Application 2

The second dataset for our research, originating from Yang [30], is presented in Table 4. The frequent occurrence of accidents at intersections in the area increases the probability of traffic accidents due to the vehicle type, traffic volume, and linear congestion. People can analyze data to discover the changes and distribution patterns of traffic volume over time and space. These findings help develop effective traffic control measures and reduce accident rates under different traffic conditions.

The investigation was conducted at the T-shaped intersection of West Ring Road outside the University City and Xingguang Road under the University City in Guangzhou Province. The survey focuses on the traffic volume of motor vehicles at this intersection, which was collected through manual monitoring.

According to Table 4, we can calculate from the original data that λ=1.911 and ϕ=3/5=0.6. Substituting these values into the maximum likelihood estimation, as shown in Equations (5) and (6), we have ${\hat{\lambda }}_{MLE}=2.101127$ and ${\hat{\phi }}_{MLE}=0.8524065$. Utilizing linear Bayesian estimation (13), we obtain ${\hat{\lambda }}_{LB}=1.817069$ and ${\hat{\phi }}_{LB}=0.852164$. Therefore, similarly to the previous example, we can obtain $MSE({\hat{\theta }}_{MLE})=0.7400561$, $MSE({\hat{\theta }}_{LB})=$0.4422734, and $MSE({\hat{\theta }}_{LB})\leq MSE({\hat{\theta }}_{MLE})$. This still indicates that the linear Bayesian estimation is superior to the maximum likelihood estimation.

## 5. Conclusions

This research study primarily focuses on an in-depth examination of the unique characteristics and distinct properties of linear Bayesian estimation, specifically applied to the misrecorded Poisson distribution. A key aspect of this study is providing robust empirical evidence that demonstrates the enhanced performance of linear Bayesian estimation when compared to maximum likelihood estimation, particularly within the framework of this distribution. Moreover, to reinforce these theoretical insights, the study employs a comprehensive approach, encompassing both rigorous validation through a series of detailed numerical simulations and the use of illustrative examples to further elucidate these findings. This multifaceted approach not only solidifies the theoretical underpinnings but also showcases the practical applicability of linear Bayesian estimation in real-world scenarios.

The method employed in this paper may not capture all dynamic characteristics when dealing with complex real-world situations. Due to the development of big data and advanced statistical methods and the need for more complex models in recent years, we consider extending the traditional framework of the Poisson distribution with the Hawkes process, thereby providing more effective tools and methodologies for solving practical problems.

## Figures and Tables

**Table 1 entropy-26-00062-t001:** Estimation results.

	λ	ϕ	${\hat{\lambda }}_{MLE}$	${\hat{\phi }}_{MLE}$	${\hat{\lambda }}_{LB}$	${\hat{\phi }}_{LB}$
N = 200	0.7	0.7	0.704053	0.694272	0.703608	0.694273
		0.9	0.712559	0.893031	0.711961	0.893031
	0.9	0.7	0.89567	0.692927	0.895587	0.692927
		0.9	0.897774	0.89222	0.897602	0.89222
N = 800	0.7	0.7	0.702591	0.699152	0.702557	0.699152
		0.9	0.69286	0.89726	0.692883	0.89726
	0.3	0.8	0.294824	0.795203	0.294818	0.795203
	0.8	0.3	0.794149	0.294536	0.794162	0.294536

**Table 2 entropy-26-00062-t002:** The MSE and distance from the true value.

	λ	ϕ	${\hat{\theta }}_{MLEMSE}$	${\hat{\theta }}_{LBMSE}$	${\left\|{\hat{\theta }}_{MLEMSE}-{\hat{\theta }}_{LBMSE}\right\|}_{2}$
N = 200	0.7	0.7	0.007665	0.007364	0.001971974
		0.9	0.00576	0.005474	0.002007057
	0.9	0.7	0.005215	0.005035	0.001461555
		0.9	0.006006	0.005752	0.001724022
N = 800	0.7	0.7	0.00149	0.001474	0.000222702
		0.9	0.001428	0.001411	0.000225155
	0.3	0.8	0.001506	0.001492	0.000243135
	0.8	0.3	0.003181	0.003162	0.000233711

**Table 3 entropy-26-00062-t003:** Soldiers who were kicked to death by horses between 1875 and 1984.

Number per Deaths of Army Corps per Year	Number of Observations	
	Original Data	Altered Data
0	109	129
1	65	45
2	22	22
3	3	3
4	1	1
5	0	0

**Table 4 entropy-26-00062-t004:** Traffic volume at intersection C.

The Number of Passes for Small-Sized Cars	Number of Observations	
	Original Data	Altered Data
0	15	18
1	5	2
2	9	9
3	6	6
4	7	7
5	1	1
6	2	2

## Data Availability

The data presented in this study are available on request from the corresponding author.

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
