# Peer review of "Linear Bayesian Estimation of Misrecorded Poisson Distribution"

_entropy, 2024, doi:10.3390/e26010062_

Round 1
Reviewer 1 Report
Comments and Suggestions for Authors
The paper presents a novel approach to parameter estimation within the statistical framework of Misrecorded Poisson Distributions by introducing a linear Bayesian estimation method. The authors highlight the importance of parameter estimation in statistical inference and address the challenges in count data analysis where misrecordings can occur. Their proposed linear Bayesian estimation method utilizes prior information and avoids cumbersome calculations, aiming to improve both accuracy and stability in parameter estimation.
The main findings of the paper suggest that the linear Bayesian method offers a more efficient solution compared to traditional methods, as confirmed by numerical simulations and examples provided. This approach seems to be particularly advantageous when dealing with misrecorded data, a common issue in real-world datasets where certain values are prone to be incorrectly recorded due to various reasons.
However, the model's simplicity raises some concerns. It could be perceived as outdated. In the context of modern statistical methodologies, the authors might benefit from considering the Hawkes process, which represents a contemporary extension of the Poisson process and could potentially provide a more sophisticated framework for their study. The Hawkes process is known for its ability to model self-exciting temporal events, which could introduce a more dynamic and complex structure to the analysis of count data, see
[1]https://cran.r-project.org/web/packages/hawkesbow/readme/README.html,
[2]https://www.frontiersin.org/articles/10.3389/fphy.2022.1019380/full,
[3]https://proceedings.mlr.press/v119/zhang20q.html,https://journals.plos.org/plosone/article?id=10.1371/journal.pone.0281370.
Furthermore, the empirical example used in section 4.2 of the paper, involving data from the 19th century about Prussian soldiers, though classic, may not resonate with current datasets and contemporary issues. A more current and relatable dataset would not only make the paper more relevant but also demonstrate the applicability of the proposed method to modern-day problems. Generalizing the paper's setting to accommodate the "Misrecorded Hawkes process" might present a more challenging and insightful extension, ensuring the paper's relevance and application to current statistical challenges.
In summary, while the paper makes a significant contribution to the field of misrecorded data analysis, there is a clear opportunity for the authors to expand their research to include more advanced processes like the Hawkes process and to apply their methods to more current and compelling datasets. This would greatly enhance the applicability and contemporary relevance of their work.
Reviewer 2 Report
Comments and Suggestions for Authors
This paper introduces a linear Bayesian estimation approach to enhance the accuracy of parameter estimation in the Misrecorded Poisson distribution, addressing the challenge of modeling count data with erroneous records. The proposed method incorporates prior information while streamlining the calculation of posterior expectations. It provides an explicit solution for linear Bayesian estimation, demonstrating its effectiveness through numerical simulations and practical examples. However, there are a few places the writing confused the readers. I would suggest major changes or rephrase of them.
Introduction: The second paragraph of this section compared the traditional methods introduced in the references with the method that will be introduced in this article without really introducing it at least in general. This made the reading hard to follow.
Methods and Results: The proof process is suggested to be put in the appendix. It is not clear that what aspects of procs and cons you evaluate. The words ‘pros and cons’ were a very general and non-informative description. Please identify the specific aspects you evaluate. Please use capital J for J times of simulation and report the value of J.
Comments on the Quality of English LanguageThe writing and the structure of organizing the paper should be improved such that it can be read more logical.
Reviewer 3 Report
Comments and Suggestions for Authors
The authors propose a linear Bayesian estimator for estimating parameters of Misrecorded Poisson distribution. The topic may not be new, but it proposes a new way of estimating parameters.
The proposed estimator performs better than the existing ML estimators.
No improvement is needed in the present work. Improvements can be dealt with in a separate paper.
The conclusions are consistent with the evidence and arguments presented.
1. The following related paper may be added to the references;
S.C. Bagui and K.L. Mehra, "The Poisson Distribution and Its Convergence to the Normal Distribution," International Journal of Statistical Science, Vl. 36, no. 5, pp. 37-56.
This paper gives a historical background for Poisson distribution and also describes some of its applications in the early days and more. Additionally, it introduces a new ratio method that demonstrates convergence to the normal distribution, which may get the researcher's attention.
2. In some cases, equations are not displayed correctly. Make sure all equations are shown in the middle of the pages.
2. keep equation numbers to the far right of the page.
Round 2
Reviewer 1 Report
Comments and Suggestions for Authors
The article titled "Linear Bayesian Estimation of Misrecorded Poisson Distribution" presents a significant contribution to the field of statistical inference, particularly in the realm of parameter estimation. The authors propose a linear Bayesian estimation methodology for the Misrecorded Poisson distribution, a valuable addition given the frequent occurrence of such data in various fields. This approach not only embraces prior information but also circumvents the cumbersome calculations typically associated with posterior expectations, which could be highly beneficial for practitioners.
Commendably, the paper provides an explicit solution for linear Bayesian estimation, showcasing its superiority through comprehensive numerical simulations and illustrative examples. The inclusion of Monte Carlo integration techniques for obtaining posterior distributions adds to the robustness and practicality of the method.
The research addresses an important gap in statistical quality control procedures, by factoring in the real-world scenario of misrecorded data, which is often overlooked. This reflects a deep understanding of the challenges faced during the data collection process, especially when accurately classifying defects in manufacturing or service industries.
The paper's strength lies in its clear structure, beginning with a thorough introduction that sets the stage for the subsequent advanced discussions. It successfully synthesizes findings from various studies on Misrecorded Poisson distributions, thereby situating the current research within a broader context of existing literature.
The authors' ability to distill complex statistical concepts into accessible language is noteworthy, making the article a potential reference for both new and seasoned researchers. The innovative nature of the research could potentially pave the way for new methodologies in handling similar statistical distributions.
However, the paper does not shy away from acknowledging its limitations, particularly regarding the dynamic characteristics of real-world data. This level of academic humility is commendable and suggests a pathway for future research.
In conclusion, this study makes a compelling case for the linear Bayesian estimation method in the context of Misrecorded Poisson distribution. It stands as a testament to the authors' meticulous research and their dedication to enhancing statistical methods for practical applications, offering a substantial contribution to the field that warrants further exploration and application.
Reviewer 2 Report
Comments and Suggestions for Authors
The revision looks good and I am satisfied with the current draft.
Comments on the Quality of English LanguageOverall it is well written.
Author Response
Thank you for your recognition of the quality of the article, thank you for your evaluation